# Unveiling Metabolic Crosstalk: *Bacillus*-Mediated Defense Priming in Pine Needles Against Pathogen Infection

**DOI:** 10.3390/metabo14120646

**Published:** 2024-11-21

**Authors:** Quan Yang, Anqi Niu, Shuang Li, Junang Liu, Guoying Zhou

**Affiliations:** 1Key Laboratory of National Forestry and Grassland Administration on Control of Artificial Forest Diseases and Pests in South China, Central South University of Forestry and Technology, Changsha 410004, China; ywuqun@126.com (Q.Y.); anqi_27a@163.com (A.N.); shuanglil@163.com (S.L.); 2Hunan Provincial Key Laboratory for Control of Forest Diseases and Pests, Central South University of Forestry and Technology, Changsha 410004, China; 3Key Laboratory of Cultivation and Protection for Non-Wood Forest Trees, Central South University of Forestry and Technology, Changsha 410004, China; 4College of Forestry, Central South University of Forestry and Technology, Changsha 410004, China; 5College of Life and Environmental Sciences, Central South University of Forestry and Technology, Changsha 410004, China

**Keywords:** *Bacillus*, metabolic programming, systemic acquired resistance, induced systemic resistance, forestry biocontrol

## Abstract

**Background/Objectives:** Plant growth-promoting rhizobacteria (PGPR), particularly *Bacillus* spp., are pivotal in enhancing plant defense mechanisms against pathogens. This study aims to investigate the metabolic reprogramming of pine needles induced by *Bacillus* csuftcsp75 in response to the pathogen *Diplodia pinea* P9, evaluating its potential as a sustainable biocontrol agent. **Methods:** Using liquid chromatography–mass spectrometry (LC-MS/MS), we performed a principal component analysis and a cluster analysis to assess the metabolic alterations in treated versus control groups. This study focused on specific metabolites associated with plant defense. **Results:** Our findings indicate that treatment with *Bacillus* csuftcsp75 significantly modifies the metabolic profiles of pine needles, leading to notable increases in metabolites associated with flavonoid biosynthesis, particularly phenylpropanoid metabolism, as well as amino acid metabolism pathways. These metabolic changes indicate enhanced systemic acquired resistance (SAR) and induced systemic resistance (ISR), with treated plants exhibiting elevated levels of defense-related compounds such as 5-hydroxytryptophol and oleanolic acid. **Conclusions:** This study reveals that *Bacillus* csuftcsp75 enhances defense against pathogen P9 by modulating pine needle metabolism and activating key immune pathways, inducing systemic acquired resistance and induced systemic resistance, offering a natural alternative to chemical pesticides in sustainable agriculture.

## 1. Introduction

In the fields of sustainable agriculture and forest management, understanding the intricate relationships between plants and their microbial allies is crucial, particularly regarding plant growth-promoting rhizobacteria (PGPR) [1,2]. The health of the soil environment, including its physical structure, chemical composition, and microbial community organization, ultimately influences plant growth [3]. Rhizosphere ecology elucidates the intricate dynamics of species interactions by examining the structure of rhizosphere microbial communities and the composition of rhizosphere exudates, thereby revealing the significance of material exchange in shaping these interactions.

The rhizosphere is a dynamic interface where complex interactions between plant roots and soil microbes occur [4]. Here, plants not only absorb nutrients but also secrete a variety of chemical signals that shape microbial communities [5]. Among these microbes, PGPR, such as *Bacillus subtilis*, play a key role in enhancing nutrient uptake, promoting plant growth, and regulating plant immune responses [6]. These bacteria form a symbiotic relationship with their host plants: they provide beneficial services such as nutrient mobilization, pathogen suppression, and the induction of systemic resistance, while plant roots secrete sugars and other organic compounds that serve as food and a habitat for the bacteria [7]. During this process, certain PGPR, including species of *B. subtilis*, can enter plant tissues through the roots and colonize the plant. This colonization, which involves a series of biochemical and physiological processes, directly impacts plant growth and immunity [8]. Studies [9,10] have shown that, compared with non-colonizing strains, PGPR colonization establishes a sustained microbial presence within the plant, effectively contributing to the regulation of its immune system [11]. Research on the interactions between *B. subtilis* and plant roots has demonstrated that these interactions lead to changes in both primary and secondary metabolites in plants. This metabolic reprogramming not only adjusts plant growth and development but also strengthens plant defenses against biotic stress [12].

Additionally, in a study on *Bacillus*-induced defense responses in chili plants, Hwe-Su et al. found that *B. subtilis* colonization of a plant produces specific compounds, such as lipopeptides, which influence the signaling pathways of systemic acquired resistance (SAR) and induced systemic resistance (ISR). This enhances the ability of plants to defend against subsequent pathogen attacks [13]. Antimicrobial compounds secreted by *B. subtilis* and the induced metabolic changes within plants create a heightened state of alert against potential pathogenic threats [14].

However, the application of PGPR in forestry faces greater challenges compared with agriculture due to the complexity of forest ecosystems and the long growth cycles of trees [15]. While the use of PGPR in agriculture primarily focuses on promoting crop growth and yield, its application in forestry is more concerned with enhancing plant resilience and facilitating environmental restoration [16]. *Pinus massoniana*, a commonly used timber species in southern China, faces significant threats from fungal diseases such as pine dieback, which are exacerbated by its dense monoculture plantations and the warm, humid climate. These conditions weaken the growth of pine forests, leading to more severe pest and disease problems. Since pine forests are often located near residential areas and farmland, the use of chemical agents must be minimized to avoid contamination, making biocontrol with PGPR a more environmentally friendly alternative for protecting tree health.

This study conducted a controlled experimental design in which pine seedlings were treated with *Bacillus* strain csuftcsp75, the pine dieback pathogen P9, or both. The responses of the seedlings were analyzed using advanced liquid chromatography–mass spectrometry (LC-MS/MS) techniques. This approach identifies key metabolic pathways influenced by PGPR and elucidates how these pathways contribute to the activation of defense mechanisms. By integrating detailed metabolomic data with bioinformatics tools, this study aims to not only identify the key metabolites involved in defense responses but also to understand the broader implications of these microbial interactions for forest health management. These findings are expected to provide fundamental knowledge for developing innovative biocontrol strategies that harness the natural defense capabilities of PGPR.

## 2. Materials and Methods

### 2.1. The Microorganisms Used in This Study

The *Bacillus* strain csuftcsp75 (NCBI, Bethesda, MA, USA, access number PQ511142) was isolated from healthy Masson pine needles, and the *Diplodia pinea* strain P9 (NCBI access number PQ511143) was isolated from Masson pine shoot blight lesions. All strains used in this study were isolated and preserved in our laboratory.

To prepare the csuftcsp75 strain, streak plating was performed using LB agar plates (prepared by adding 1 g of NaCl, 1 g of tryptone, 0.5 g of yeast extract, and 2 g of agar to 100 mL of water; the preparation was mixed well, steam-sterilized, and poured onto the plates). The plates were incubated at 37 °C in the dark for over 16 h. Then, single colonies were picked and inoculated into liquid LB without agar to prepare a shaking flask culture (at 28 °C with 200 rpm). The obtained bacterial culture was centrifuged at 6000 rpm. The pellet was resuspended and washed twice with sterile water and then resuspended in sterile water and adjusted to a concentration of 10^9 colony-forming units per milliliter (cfus/mL). The bacterial water suspension at this concentration was used for irrigation in pot tests.

To prepare the P9 strain, pine needle PDA agar plates (prepared by boiling 15 g of potatoes and 5 g of needles from *Pinus massoniana*, filtering the broth, adding 2 g of glucose, 0.1 g of yeast extract, and 2 g of agar to the filtrate, and then adjusting the volume to 100 mL with water; the preparation was steam-sterilized and poured onto the plates) were used for block inoculation and incubated at 28 °C in the dark for over 36 h. Then, mycelium was picked and inoculated into liquid PDB (PDA without agar) for culture at 28 °C with 180 rpm. The cultures were regularly sampled to observe the spore production of the fungus, and finally, sterile lens paper was used to filter the culture. The filtrate was collected and centrifuged at 6000 rpm to harvest the spores. The pellet was resuspended and washed twice with sterile water and then resuspended in sterile water and adjusted to a concentration of 10^8^ cfus/mL. This spore suspension of P9 was used to spray the shoots of pine seedlings to create a pathosystem model for pine shoot blight in pot tests.

### 2.2. Experimental Grouping and Planting Environment

*Pinus massoniana* seedlings were ordered from the Hunan Provincial Academy of Forestry Science as S9612-Pm1a containerized seedlings, with a height of 20–25 cm. The pine seedlings were divided into 4 groups (refer to Section 2.3 for details), with 5 plants per group as biological replicates.

### 2.3. Inoculation and Sampling

The grouping and treatment conditions of the potted plant experiment are presented in Table 1.

The IT and PIT groups were irrigated with the csuftcsp75 culture prepared in Section 2.1, with 100 mL per pot each time. The plants were irrigated once a day for a total of 3 days.

An amount of 2% mannitol was added to the P9 spore solution prepared in Section 2.1, mixed well, and then the P9 spore solution was evenly sprayed onto the needle leaves of the pine seedlings in the PT and PIT groups using a sprayer, followed by sealing with a transparent plastic bag.

For all treatment groups, about 5 g of pine needles was collected from the middle part of the pine seedlings at 72 and 144 h (3 and 7 days) after the inoculation of the csuftcsp75 strain in the IT and PIT groups and immediately quenched with liquid nitrogen. The samples were then sent to Beijing Biomics Biotech Co., Ltd., Beijing, China, for non-targeted metabolomics detection.

### 2.4. Metabolite Detection and Analysis

#### 2.4.1. Metabolite Extraction

The pine needle samples were rinsed with sterile water and then subjected to vacuum freeze-drying. An extraction solvent (a mixture of methanol, acetonitrile, and water in a volume ratio of 2:2:1) was added and mixed evenly. The samples were treated with a 45 Hz grinder and sonicated in an ice water bath for 10 min. After that, they were centrifuged at 4 °C and 12,000 rpm for 15 min to collect the supernatant, followed by drying the extract using a vacuum concentrator. The dried extract was redissolved with an enrichment solution (a mixture of acetonitrile and water in a volume ratio of 1:1) and then fully redissolved a second time, centrifuged again to collect the supernatant, and finally subjected to LC-MS analysis.

The sample extracts were analyzed using a UPLC-ESI-MS/MS system (UPLC, Waters Acquity I-Class PLUS, Milford, MA, USA; MS, Sciex Applied Biosystems QTRAP 6500+, Framingham, MA, USA). The analytical conditions were as follows: UPLC: column, Waters HSS-T3 (1.8 µm, 2.1 mm × 100 mm); the mobile phase comprised solvent A, which consisted of pure water with 0.1% formic acid and 5 mM of ammonium acetate, and solvent B, which consisted of acetonitrile with 0.1% formic acid. Sample measurements were performed with a gradient program that employed the starting conditions of 98% A and 2% B and maintained for 1.5 min. Within 5.0 min, a linear gradient to 50% A and 50% B was programmed, within 9.0 min, a linear gradient to 2% A and 98% B was programmed, and a composition of 2% A and 98% B was maintained for 1 min. Subsequently, a composition of 98% A and 2% B was adjusted within 1 min and maintained for 3 min. The flow velocity was set as 0.35 mL per minute; the column oven was set to 50 °C; and the injection volume was 2 μL. The effluent was alternatively connected to an ESI–triple quadrupole–linear ion trap (QTRAP)-MS (Sciex Applied Biosystems QTRAP 6500+, Framingham, MA, USA).

#### 2.4.2. LC-MS/MS Analysis

The ESI source operation parameters were as follows: a source temperature of 550 °C; ion spray voltage (IS) of 5500 V (positive ion mode)/−4500 V (negative ion mode); the ion source gas I (GSI), gas II (GSII), and curtain gas (CUR) were set at 50, 55, and 35 psi, respectively; and the collision-activated dissociation (CAD) was medium. Instrument tuning and mass calibration were performed with 10 and 100 μmol/L polypropylene glycol solutions in QQQ and LIT modes, respectively. QQQ scans were acquired as MRM experiments with the collision gas (nitrogen) set to medium. Analysis of DP (declustering potential) and CE (collision energy) parameters for individual MRM transitions was carried out with further DP and CE optimization. A specific set of MRM transitions was monitored for each period according to the metabolites eluted within this period.

#### 2.4.3. Data Analysis

After normalizing the original peak area information with the total peak area, a follow-up analysis was performed. Principal component analysis and Spearman correlation analysis were used to assess the repeatability of the samples within the group and the quality control samples. KEGG and HMDB were used to search for classification and pathway information about the identified compounds.

### 2.5. Statistical Analyses

The difference multiples were calculated and compared according to the grouping information. A T test was used to calculate the difference significance *p* value of each compound. The R (R version 4.4.1) language package ropls was used to perform OPLS-DA (Orthogonal Partial Least Squares Discriminant Analysis) modeling, and 200 times permutation tests were performed to verify the reliability of the model. The VIP value of the model was calculated using multiple cross-validation. The method of combining the difference multiple, the *p* value, and the VIP value of the OPLS-DA model was adopted to screen the differential metabolites. The screening criteria were log2FC > 1 or log2FC < −0.5, *p* value < 0.05, and VIP > 1. Using the absolute expression levels of all metabolites as factors and the categories of treatment groups as influencing levels, cluster analysis of the expression patterns of metabolites in all groups was performed using both Euclidean distance with Ward’s linkage method and Manhattan distance with average linkage method, and a dendrogram was generated.

## 3. Results

### 3.1. Clustering and Differences in Plant Metabolic Pathways Under Different Treatments

The seedlings of pine in each treatment group and their sampling locations are shown in Figure 1. We conducted principal component analysis (PCA) (Figure 2A) and cluster analysis (Figure 2B,C) on the absolute expression levels of metabolites across all treatment groups. The specific group names and their meanings are detailed in Table 2. The results showed that after 72 h of treatment, the *Bacillus* csuftcsp75-induced group (IT1) and the induced infection group (PIT1) consistently clustered together. Additionally, in the cluster analysis, the infection group (PT1) was located on the same primary branch as the induced groups (IT1 and IT2). Based on the branch structure, differences were observed between PT1, IT2, and other treatments within the same branch. At 144 h post infection, the induction effect of csuftcsp75 on metabolic expression was not prominent in the PT2 and PIT2 groups. Therefore, we speculate that the potential impact of csuftcsp75 mainly occurs in the early stages of pathogen infection (72 h), and subsequent metabolomics analysis will focus on the comparison between CK1, IT1, and PIT1.

### 3.2. KEGG Pathway Analysis Under Early Infection and csuftcsp75 Induction Treatments

We analyzed the differentially expressed metabolites within KEGG pathways across treatment groups sampled 72 h after treatment ended (Figure 3). The findings revealed that both the csuftcsp75 induction group (IT, Figure 3B) and the pathogen infection group (PT, Figure 3A) exhibited similar changes in metabolite profiles; however, they differed significantly in the degree and types of pathway activation. Csuftcsp75 induction notably enhanced the activity of pathways related to amino acid metabolism, with a greater variety of pathways involved in the degradation and synthesis of different types of amino acids being identified in the top-ranking metabolic pathway enrichments. Additionally, in the csuftcsp75 induction group, there was upregulation of glucosinolate, monoterpenoid, and phenylpropanoid synthesis pathways. The ABC transporter pathway was also fully activated in both csuftcsp75-induced and pathogen-infected plants, indicating its significant role in response to external threats. In contrast, lipid and diterpene metabolism pathways were less active in the csuftcsp75-induced group compared to the pathogen-infected group. Specifically, the pathogen-infected group (PT) showed increased activity in glycerophospholipid metabolism and biosynthesis of cutin, suberin, and waxes, pathways closely associated with cell membrane and surface structure.

To investigate the relationship between plant metabolites and their KEGG pathways under different treatments, we conducted a principal component analysis (PCA) using differential metabolic pathways as variables and treatment groups as the analytical framework (Figure 4A). This analysis highlighted key metabolic pathways significantly regulated under the CK (control), PT (P9 treatment), and IT (*Bacillus* csuftcsp75 treatment) conditions, and the main differential metabolites in each group are listed in Table 3. Although these metabolites share pathways, their differential up- or downregulation reflects unique metabolic profiles for each treatment. Five key pathways aligned with the principal components were identified: ABC transporter (ko02010), flavonoid biosynthesis (ko00941), aminoacyl–tRNA biosynthesis (ko00970), D-amino acid metabolism (ko00470), and purine metabolism (ko00230). The heatmaps in Figure 4B–F illustrate metabolite expression across treatments, showing nuanced immune and metabolic responses in Masson pine. Notable upregulation in PT, such as for L-glutamate, suggests enhanced signaling and metabolic shifts during pathogen invasion, while inosine upregulation in IT indicates defense responses triggered by *Bacillus*.

### 3.3. Expression Analysis of Significantly Different Metabolites

We used OPLS-DA scoring to evaluate significant differences in the metabolic profiles of pine needles among the PT (pathogen P9 treatment), IT (csuftcsp75 treatment), and CK (control) groups (Figure 5A,C). OPLS-DA serves as an effective tool for identifying biomarkers across sample groups, optimizing model predictability by maximizing inter-group variation while minimizing intra-group variability. Using strict selection criteria (log2FC > 1 or < −0.5, *p* < 0.05, and VIP > 1), we identified the top 20 metabolites with the most significant differences between PT and IT relative to CK (Figure 5B,D). These metabolites, which may play essential roles in plant responses to pathogen infection and biocontrol induction, are visualized in a heatmap, with color gradients indicating relative expression levels across groups. In both the PT and IT groups, certain metabolites such as 5-hydroxytryptamine, magnoflorine, isobutyryl-L-carnitine (chloride), oleanolic acid, hesperetin, and naringin showed pronounced upregulation or downregulation, with expression changes by magnitudes of 10^2^ or more.

## 4. Discussion

### 4.1. Enhanced Defense Metabolism in Masson Pine Under Bacillus csuftcsp75 Treatment

Our findings indicate that *Bacillus* csuftcsp75 treatment induces extensive metabolic adjustments in Masson pine needles, primarily contributing to enhanced resistance against pathogen P9. Cluster analysis across different treatments and time points revealed similar metabolic shifts in both the csuftcsp75-treated (IT) and early pathogen-infected (PT) groups, suggesting that csuftcsp75 establishes a foundational defense state comparable to initial pathogen response. Nonetheless, the separation between PT1, IT2, and other groups points to unique pathways activated by csuftcsp75 that may support resilience beyond that conferred by pathogen exposure alone.

Metabolite analysis highlights a prominent upregulation of compounds within the phenylpropanoid pathway, known for its critical role in plant defense mechanisms. Elevated flavonoid levels, for example, contribute both as antioxidants and as direct antimicrobial agents, inhibiting pathogen proliferation. This aligns with Diao et al. [17], who observed that anthocyanins and luteolin produced in sorghum under pathogen stress exhibited toxicity toward *Colletotrichum* spores. Similarly, Wang et al. [18] demonstrated that increased flavonol and anthocyanin synthesis in sorghum enhances resistance to *Colletotrichum* infections. These findings reinforce our results, where the accumulation of flavonoids in csuftcsp75-treated pine needles likely bolsters defense against P9. Notably, flavonoid biosynthesis not only aids pathogen resistance but also influences immune regulation, maintaining homeostasis within the plant [19,20].

Beyond flavonoids, other secondary metabolites were also notably upregulated, consistent with their roles in pathogen defense. For instance, hydroxybenzoic acid, shown by Chen et al. [21] to interfere with pathogen metabolism, was significantly elevated in the csuftcsp75 treatment, suggesting its involvement in the pine’s defensive response. Halina et al. [22] further described hydroxybenzoic acid’s role in strengthening cellular defenses by reducing oxidative stress and enhancing cell wall integrity, supporting similar protective functions observed here. Coumarins also emerged as key metabolites, known for their influence on root microbial communities under stress [23] and their antimicrobial effects during pathogen encounters [24], both of which could contribute to a fortified rhizosphere in csuftcsp75-treated plants.

### 4.2. Distinct Defense Mechanisms in Pathogen-Induced vs. Biocontrol-Agent-Induced Responses

The differences in the metabolite expression patterns between the PT and IT groups indicate that csuftcsp75 and pathogen P9 activate distinct, adaptive defense mechanisms in Masson pine. The ABC transporter pathway likely supports the enhanced transport of defense compounds and signaling molecules. Upregulated flavonoid biosynthesis provides antioxidant and antimicrobial protection, while aminoacyl–tRNA biosynthesis may support protein synthesis necessary for sustained stress response. D-amino acid metabolism may also facilitate plant–microbe interactions by serving as signaling or antimicrobial precursors. Notably, purine metabolism, which is linked to energy production and signaling through derivatives like adenosine, suggests elevated defense readiness [25].

Metabolites such as 5-hydroxytryptamine, magnoflorine, and isobutyryl-L-carnitine (chloride) were upregulated in both treatment groups, implying roles in modulating defense signaling and energy metabolism. While PT samples showed increased metabolites associated with immediate pathogen-triggered defenses, IT samples contained higher levels of polyols and polyhydric alcohols, indicating an antioxidant-driven protective response induced by csuftcsp75. This suggests that pathogen infection provokes a targeted, rapid defense response, while csuftcsp75 stimulates a more generalized, sustained antioxidant and immunity-boosting effect in Masson pine.

Additionally, elevated levels of salicylic acid and jasmonic acid further support the activation of immune pathways by csuftcsp75. These signaling molecules, commonly induced by plant growth-promoting rhizobacteria (PGPR) like csuftcsp75, play essential roles in both local and systemic immune responses [26], preparing the plant for potential pathogen challenges. This broad immune activation enhances defense mechanisms, allowing for a faster, more coordinated reaction upon subsequent pathogen exposure.

### 4.3. Mechanisms of Local and Systemic Immune Induction by Bacillus csuftcsp75

*Bacillus* csuftcsp75 induces plant immune responses through multiple mechanisms, involving both local and systemic defense mechanisms. The evidence for enhanced local defense responses comes from the accumulation of defensive metabolites at the sites of bacterial colonization. Our metabolic analysis indicates a significant upregulation of defense-related metabolites in these areas, suggesting an enhancement in the plant’s local defense mechanisms [27]. This local response may directly activate defense genes via inducer–receptor interactions, which is consistent with the findings of Jelena et al., who revealed how microbe-associated molecular patterns trigger localized immune responses through the recognition of microbial metabolites [28].

Endophytic bacteria enhance plant defense capabilities through various means. In addition to directly inhibiting pathogen growth by secreting antimicrobial substances, antagonistic microbes can improve plant resistance by activating induced systemic resistance (ISR) [29]. Studies have shown that endophytic bacteria can release oligosaccharides from plant cell walls by secreting enzymes that degrade host cell walls, such as the α-mannosidase ShAM1 studied by Bai et al. [30], thereby activating DAMP (damage-associated molecular pattern)-triggered immune responses.

### 4.4. Systemic Metabolic Reprogramming and Defense Optimization

In addition to localized responses, *Bacillus* csuftcsp75 induces a wide-ranging metabolic reprogramming that enhances systemic immunity in Masson pine. Our study observed a significant upregulation in amino acid metabolism, glucosinolates, monoterpenoids, and phenylpropanoids across various tissues, compounds known to reinforce antimicrobial defense and cell wall integrity. This systemic metabolic adjustment aligns with Wilson et al.’s work, which demonstrates that plants transmit stress signals between organs to activate broader immune responses [31]. Our results indicate that csuftcsp75 primes the plant for a heightened state of readiness by modulating key pathways involved in both growth and defense.

While csuftcsp75 and pathogen P9 both activate similar pathways, their impact on resource allocation differs. In the csuftcsp75-treated group, our study found a restrained activation of lipid and diterpene metabolism, likely reflecting a defense strategy that conserves resources while maintaining immunity. Conversely, pathogen P9 triggers significant activation of lipid metabolism pathways, particularly glycerophospholipid biosynthesis and cutin production, directly supporting defense against pathogenic invasion. This differential response suggests that while P9 provokes a rapid, resource-intensive defense, csuftcsp75 induces a balanced, sustained immunity that could support both growth and resilience.

Our findings also highlight the role of signaling molecules like salicylic acid in systemic immune responses, consistent with Wu et al.’s demonstration of salicylic acid’s role in preparing plants for pathogen challenges [32]. Our data support this, showing that csuftcsp75 treatment results in salicylic acid accumulation, enabling plants to mount more effective and rapid defenses upon pathogen attack.

Further supporting systemic defense, Rakhmatullina et al. and Resna et al. demonstrated how antagonistic microbes influence the transcriptional reprogramming of immune response genes through complex signaling networks, optimizing plant resource allocation for sustained immune readiness [33,34]. This is corroborated by our findings, which suggest that csuftcsp75 optimizes immune capacity while minimizing metabolic costs. Additionally, the interaction between csuftcsp75 metabolites and plant metabolites enhances plant defense sustainability. Radhakrishnan et al. observed that microbial metabolites integrate with plant metabolic pathways to improve disease resistance and resilience, with applications for sustainable agriculture and forestry [35]. This was supported by our findings, indicating that csuftcsp75 enhances both disease resistance and overall plant health, facilitating sustainable plant management [36].

In conclusion, *Bacillus* csuftcsp75 significantly strengthens Masson pine’s immune defenses through a coordinated activation of localized and systemic responses. Our study highlights the potential of csuftcsp75 as a biocontrol agent with applications in sustainable agriculture and forestry. This microbe-based strategy provides a promising approach to disease management, laying the groundwork for further studies on the interactions between microbial activity and plant metabolic processes.

## 5. Conclusions

This study shows that *Bacillus* csuftcsp75 significantly modulates pine needle metabolism, strengthening defenses against pathogen P9. Principal component and cluster analyses revealed distinct metabolic profiles in *Bacillus*-treated groups (IT and PIT) compared with the control and pathogen-only groups (CK and PT), suggesting that csuftcsp75 stabilizes metabolic functions even under pathogen stress, indicating a strong immunomodulatory effect.

A KEGG pathway analysis further indicated that *Bacillus* csuftcsp75 activates key immunity-related pathways, including flavonoid biosynthesis, amino acid metabolism, and tRNA synthesis, which are essential for both direct defense and immune signaling. This reprogramming allows rapid adaptation to environmental stressors, enhancing resilience against infections.

The differential metabolite expression between the treated and control groups supports the role of *Bacillus* csuftcsp75 in inducing systemic acquired resistance (SAR) and induced systemic resistance (ISR), as observed in the upregulation of defense-related metabolites such as 5-hydroxytryptophol and oleanolic acid. These findings underscore the potential of microbial inoculants in sustainable agriculture, offering a natural alternative to chemical pesticides. Further research into the molecular interactions of csuftcsp75 with pine defense mechanisms will advance disease management strategies in forestry and agriculture.

## Figures and Tables

**Figure 1 metabolites-14-00646-f001:**
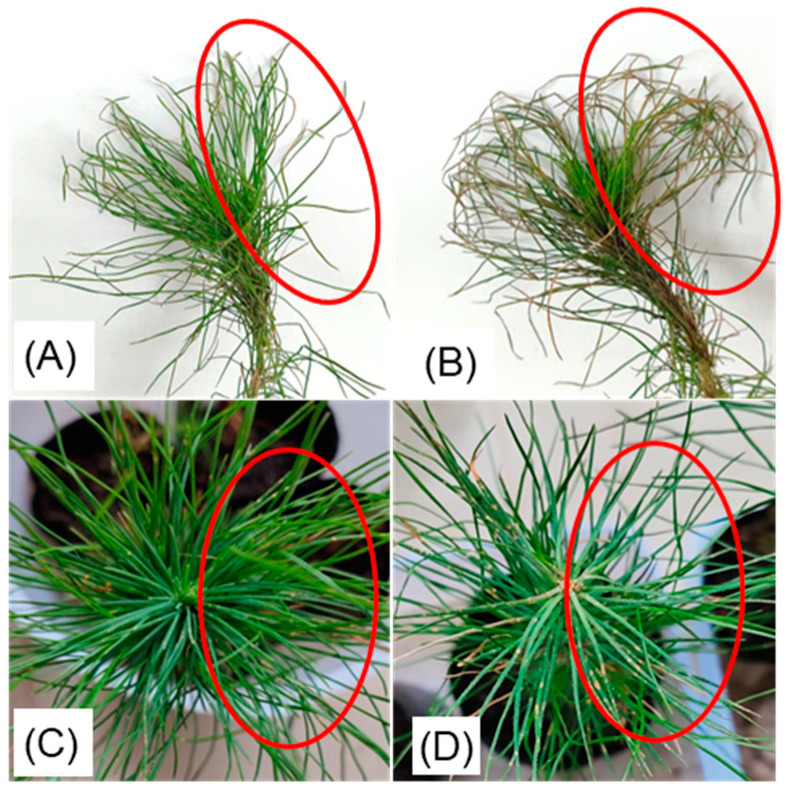
Pine seedlings after various treatments. The red circles in the figures indicate the sampling areas of the pine needles, from which we randomly collected pine needles until we obtained the appropriate weight. Another batch was collected from the same areas after 144 h of treatment. (**A**) CK group pine seedlings; (**B**) PT group pine seedlings 72 h after treatment; (**C**) IT group pine seedlings 72 h after treatment; and (**D**) PIT group pine seedlings 72 h after treatment.

**Figure 2 metabolites-14-00646-f002:**
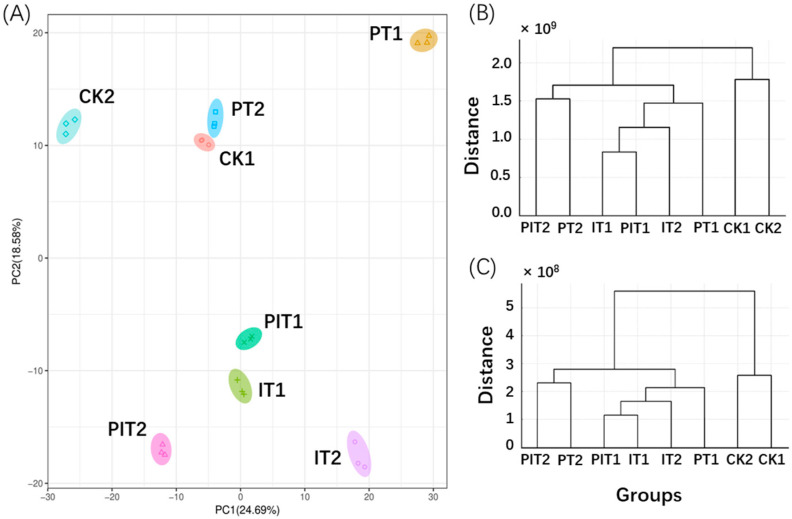
Principal component analysis and clustering analysis of the expression levels of metabolites in each treatment group. (**A**) Results of the principal component analysis of the expression levels of metabolites in each treatment group; (**B**) dendrogram of the expression of metabolites in each group calculated using Euclidean distance and Ward’s linkage; and (**C**) dendrogram of the expression of metabolites in each group calculated using Manhattan distance and average linkage.

**Figure 3 metabolites-14-00646-f003:**
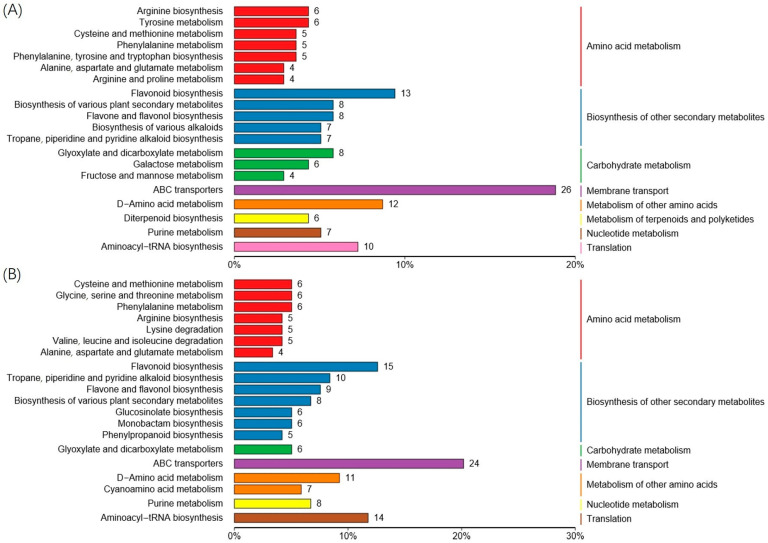
KEGG pathway enrichment results of differential metabolites in Masson pine produced by the PI and IT groups relative to the blank control after 72 h of treatment. The horizontal axis is the percentage of the number of pathways out of the total number of differential pathways, and the vertical axis is the name of the metabolic pathway: (**A**) top 20 metabolic changes in Masson pine caused by pathogen infection (PI) after 3 days of treatment; (**B**) top 20 metabolic changes in Masson pine caused by csuftcsp75 induction (IT) after 3 days of treatment.

**Figure 4 metabolites-14-00646-f004:**
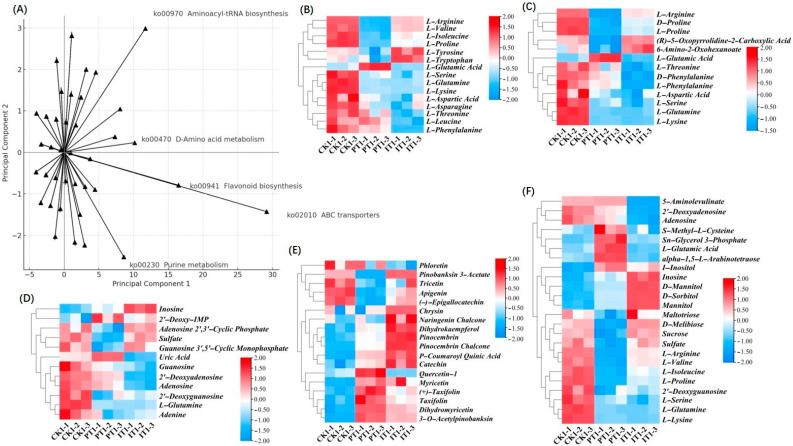
Significant analysis of metabolites in CK vs. PT and CK vs. IT groups. (**A**) Two-dimensional diagram of the correlation between metabolic pathways and treatment; the closer the variable is to the axis, the stronger the correlation of the variable with that direction; the longer the line, the higher the weight of the variable on the principal component. (**B**) Heatmap of the expression of significant metabolites in ko00970 (aminoacyl–tRNA biosynthesis) in each group. (**C**) Heatmap of the expression of significant metabolites in ko00470 (D-amino acid metabolism) in each group. (**D**) Heatmap of the expression of significant metabolites in ko00230 (purine metabolism) in each group. (**E**) Heatmap of the expression of significant metabolites in ko00941 (flavonoid biosynthesis) in each group. (**F**) Heatmap of the expression of significant metabolites in ko02010 (ABC transporters).

**Figure 5 metabolites-14-00646-f005:**
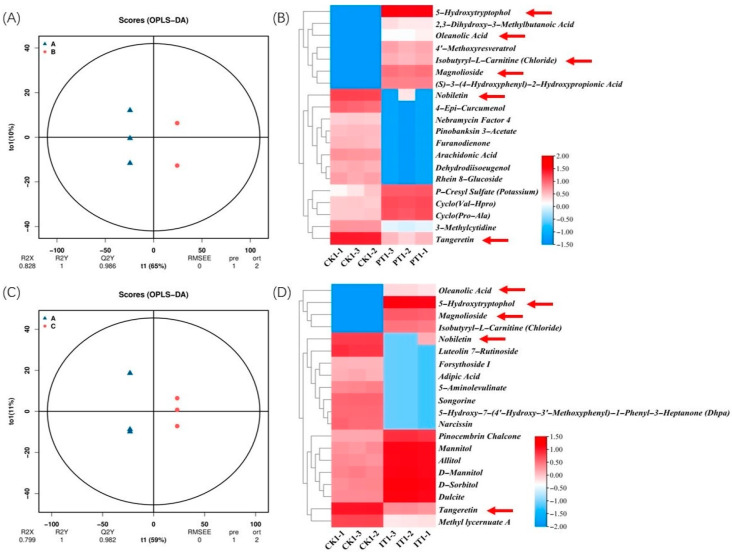
Significant analysis of metabolites in CK vs. PT and CK vs. IT groups. The legend is the log2FC value, and the red arrow indicates the same metabolites in the differential metabolites of the two groups. (**A**) OPLS-DA score of differential metabolites in Masson pine produced by pathogen infection (PI) relative to the blank control after 3 days of treatment. (**B**) Heatmap of the top 20 (Log2FC) relative expression metabolites produced by pathogen infection (PI) relative to the blank control after 3 days of treatment in Masson pine. (**C**) OPLS-DA score of differential metabolites in Masson pine produced by csuftcsp75 induction (IT) relative to the blank control after 3 days of treatment. (**D**) Heatmap of the top 20 (Log2FC) relative expression metabolites produced by csuftcsp75 induction (IT) relative to the blank control after 3 days of treatment in Masson pine.

**Table 1 metabolites-14-00646-t001:** Grouping and treatments in the potted plant experiment.

Group Name	Abbreviation	Treatments	Biological Replicates
Control group	CK	No additional treatments were applied apart from the application of sterile distilled water once a day for a total of 3 days (100 mL per pot).	5 pots (one pine seedling per pot)
Pathogen-infected group	PT	The P9 spore solution was evenly sprayed onto the needle leaves of the pine using a sprayer followed by sealing with a transparent plastic bag. The plants were irrigated with sterile distilled water once a day for a total of 3 days (100 mL per pot).	5 pots (one pine seedling per pot)
*Bacillus*-infected group	IT	The plants were irrigated with the csuftcsp75 suspension prepared in Section 2.1, with 100 mL per pot each time. The plants were irrigated once a day for a total of 3 days.	5 pots (one pine seedling per pot)
Pathogen- and *Bacillus*-infected group	PIT	The P9 spore solution was evenly sprayed onto the needle leaves of the pine using a sprayer followed by sealing with a transparent plastic bag. The plants were irrigated with the csuftcsp75 suspension prepared in Section 2.1, with 100 mL per pot each time. The plants were irrigated once a day for a total of 3 days.	5 pots (one pine seedling per pot)

**Table 2 metabolites-14-00646-t002:** Descriptions of the treatment group abbreviations in Figure 1.

Group Abbreviation	Description
CK1	Control group: the sampling time point was 72 h post treatment.
CK2	Control group: the sampling time point was 144 h post treatment.
PT1	Pathogen-infected group: the sampling time point was 72 h post treatment.
PT2	Pathogen-infected group: the sampling time point was 144 h post treatment.
IT1	*Bacillus*-infected group: the sampling time point was 72 h post treatment.
IT2	*Bacillus*-infected group: the sampling time point was 144 h post treatment.
PIT1	Pathogen- and *Bacillus*-infected group: the sampling time point was 72 h post treatment.
PIT2	Pathogen- and *Bacillus*-infected group: the sampling time point was 144 h post treatment.

**Table 3 metabolites-14-00646-t003:** The main specific metabolites induced by treatments in each group.

The Main Specific Metabolites	
IT (vs. CK)	PT (vs. CK)
(R)-5-oxopyrrolidine-2-carboxylic acid	(−)-Epigallocatechin
2′-deoxyguanosine	(+)-Taxifolin
6-amino-2-oxohexanoate	2′-deoxyadenosine
Catechin	2′-deoxy-IMP
Chrysin	3-O-acetylpinobanksin
Dihydrokaempferol	5-aminolevulinate
D-mannitol	Adenine
D-melibiose	Adenosine
D-phenylalanine	Adenosine 2′,3′-cyclic phosphate
D-proline	Alpha-1,5-L-arabinotetraose
D-sorbitol	Apigenin
I-inositol	Dihydromyricetin
Inosine	Guanosine
L-arginine	Guanosine 3′,5′-cyclic monophosphate
L-asparagine	L-arginine
L-aspartic acid	L-glutamic acid
L-glutamine	L-glutamine
L-isoleucine	L-isoleucine
L-leucine	L-lysine
L-lysine	L-proline
L-phenylalanine	L-serine
L-proline	L-threonine
L-tryptophan	L-valine
L-tyrosine	Maltotriose
L-valine	Myricetin
Mannitol	Quercetin-1
Naringenin chalcone	S-methyl-L-cysteine
P-Coumaroyl quinic acid	Sn-glycerol 3-phosphate
Phloretin	Sucrose
Pinobanksin 3-acetate	Sulfate
Pinocembrin	Taxifolin
Pinocembrin chalcone	Tricetin
Sulfate	Uric acid

## Data Availability

The plant metabolomic data used in this study have been uploaded to the MetaboLights database with the accession number MTBLS11136. The data can be found at the following URL: www.ebi.ac.uk/metabolights/MTBLS11136 (accessed on 21 September 2024).

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
