# Peer review of "Unveiling Metabolic Crosstalk: Bacillus-Mediated Defense Priming in Pine Needles Against Pathogen Infection"

_metabolites, 2024, doi:10.3390/metabo14120646_

Round 1

Reviewer 1 Report

Comments and Suggestions for Authors

The study of changes in plant metabolomics under the influence of various biotic factors is one of the actively developing and researched areas of biological research. The authors paid attention to the complex effect of rhizobacteria and pathogen on the metabolism of Pinus seedling leaves. This direction is interesting for the reader and relevant. However, it is impossible to assess the quality and significance of the material presented in the article, since there is no data on the production of extracts and their preparation. 

The methodology does not describe the control variant - what was used in this case. The captions under the figures do not contain information on the studied variants. The Results section contains a lot of text discussing the data, which is better presented in the Discussion section.

Conclusion: The work requires significant revision.

Comments on the Quality of English Language

ThThe text and its presentation in English need to be revised.e text and its presentation in English need to be revised.

Author Response

Comment1: The methodology does not describe the control variant - what was used in this case.
Response1: We have rechecked the description of the groups and design, and it is true that the previous version omitted this necessary explanation. The detailed group description and corresponding treatments can now be found in Table 1 of Section 2.3, and we hope it adequately addresses the issue.

Comment2: The captions under the figures do not contain information on the studied variants.
Response2: Thank you for pointing this out. Upon review, the figure captions in Section 3.1 indeed lacked clarity in explanation. Considering the numerous labels and names, we have added Table 2 to display the information in Figure 2 (originally Figure 1). We hope the current presentation meets the standards.

Comment3: The Results section contains a lot of text discussing the data, which is better presented in the Discussion section.
Response3: We acknowledge the reviewer's point on this matter. We always attempt to add some explanation after presenting the data to illustrate its significance. Currently, we have moved most of the discussion-type text to the discussion section and reorganized it, hoping that the current version meets the standards.

Comment4: The text and its presentation in English need to be revised.
Response4: We have requested a fast review from MDPI, and the current version is one that has undergone language modification. We hope that the language level no longer poses a barrier to understanding.

Reviewer 2 Report

Comments and Suggestions for Authors

the study is interesting and results are discussion are the main backbone of manuscript which are written nicely. but try to focus on material and method so as to maintain its replicability. the protocols must be described in detail.

1. explain the full form of all the abbreviations for the first time. though some of the abbreviations are explained later in page 7 but it must come earlier

2. what is the history of Csuft 75 and P9 isolate, where they were isolated from and their other informations are missing. the authors have directly used these terms without mentioning any details of them

3. what is It and PIT, the treatments are also not clear, either make them clear in text itself or making a table will help the readers to under the treatments better. 

4. it is also not clear how many replications were used for each treatments

Author Response

Comment 1: Explain the full form of all the abbreviations for the first time. Although some of the abbreviations are explained later on page 7, they must be presented earlier.
Response 1: Thank you for pointing this out; you can now find the full term for OPLS-DA added in lines 184-185.

Comment 2: What is the history of Csuft 75 and P9 isolate, where they were isolated from, and their other information is missing. The authors have directly used these terms without mentioning any details about them.
Response 2: These two microbial strains were isolated from pine needles in our laboratory, one being an endophytic bacterium isolated from healthy pine needles and the other a pathogenic fungus isolated from diseased spots. We have now added a brief description and uploaded the sequences used for species identification to the GenBank database, can be found in line 92-94. hoping this clarifies the matter.

Comment 3: What is IT and PIT, the treatments are also not clear; either make them clear in the text itself or creating a table will help readers understand the treatments better.
Response 3: I think your suggestion is indeed very useful. We have now added Table 1 in Section 2.3, where we have explained our groups and treatment methods as clearly as possible. Hopefully, your questions have been answered.

Comment 4: It is also not clear how many replications were used for each treatment.
Response 4: Yes, that was an unfortunate oversight. You can now find the information in Table 1 of Section 2.3. Thank you for your correction.

Reviewer 3 Report

Comments and Suggestions for Authors

Yang et al. reported metabolic alterations in pine needles mediated by a plant growth-promoting rhizobacterium (PGPR), Bacillus csuft75. Below are some queries and comments:

1.      The authors mention: “The Bacillus strain csuft75 and Diplodia pinea strain P9 used in this study were isolated and preserved in our laboratory.” Please provide references or detailed information regarding the isolation, identification process, and GenBank accession numbers, if available.

2.      Line 103: Please spell out abbreviations (IT, PIT, PT) the first time they are used. Later in the results, terms such as PT1, PT2, IT1 are introduced—please clarify these abbreviations.

3.      Section 2.3: This section lacks clarity on the experimental design. Please specify how treatments were conducted, the number of replicates, experimental repeats, and details of plant and bacterial controls.

4.      The text states: “The IT group and PIT group were irrigated with the csuft75 culture prepared in 2.1, with 100 mL per pot each time.” Please clarify whether the bacteria were grown in an LB medium or suspended in water for irrigation.

  1. Please provide details on how the LC-MS samples were prepared after treatment. Were the needles washed or surface-sterilized prior to analysis?
  2. It would be helpful to include images of both treated and untreated plants, with clear labeling of the plant parts from which needles were collected.
  3. Before conducting PCA, it would be useful to include a table highlighting unique or major metabolites induced in the treated group to give a clearer understanding of the key metabolic changes.
  4. The introduction requires rewriting for better coherence, as it currently reads more like a literature review. For example, the sentence “This introduction outlines the…” could be restructured for smoother flow.
  5. In some places Bacillus are italicised while in some places not
Comments on the Quality of English Language

The introduction requires rewriting for better coherence, as it currently reads more like a literature review. For example, the sentence “This introduction outlines the…” could be restructured for smoother flow.

In some places Bacillus are italicised while in some places not

Author Response

Comment1: The authors mention: “The Bacillus strain csuft75 and Diplodia pinea strain P9 used in this study were isolated and preserved in our laboratory.” Please provide references or detailed information regarding the isolation, identification process, and GenBank accession numbers, if available.
Response1: Yes, we have added the relevant information along with the accession numbers, which can be found at lines 92-94.

Comment2: Line 103: Please spell out abbreviations (IT, PIT, PT) the first time they are used. Later in the results, terms such as PT1, PT2, IT1 are introduced—please clarify these abbreviations.
Response2: Yes, this is crucial. We have added Table 1 in Section 2.3 and Table 2 in Section 3.1 to display all group information, including abbreviations and full names. We hope this information thoroughly answers the question.

Comment3: Section 2.3: This section lacks clarity on the experimental design. Please specify how treatments were conducted, the number of replicates, experimental repeats, and details of plant and bacterial controls.
Response3: Agreed. We have attempted to rectify this issue by displaying each group's treatment and repetition in Table 1 in Section 2.3. We hope nothing has been overlooked.

Comment4: The text states: “The IT group and PIT group were irrigated with the csuft75 culture prepared in 2.1, with 100 mL per pot each time.” Please clarify whether the bacteria were grown in an LB medium or suspended in water for irrigation.
Response4: Agreed. The detail you pointed out should indeed be clarified. We have now added specific information, which you can find at lines 102-105, lines 114-116, and in Table 1 of Section 2.3 regarding the specific components of the suspension, preparation methods, and usage.

Comment5: Please provide details on how the LC-MS samples were prepared after treatment. Were the needles washed or surface-sterilized prior to analysis?
Response5: Thank you for pointing this out; we indeed overlooked the description of these details. In summary, the sample preparation and extraction methods can now be found in Section 2.4.1, lines 142-150.

Comment6: It would be helpful to include images of both treated and untreated plants, with clear labeling of the plant parts from which needles were collected.
Response6: Perhaps yes. We have attempted to provide some images, which are now combined as Figure 1 and can be found in Section 3.1. We hope this figure addresses your concerns.

Comment7: Before conducting PCA, it would be useful to include a table highlighting unique or major metabolites induced in the treated group to give a clearer understanding of the key metabolic changes.
Response7: Although we believe the heatmap already indicates sufficient metabolite information, we still respect and value your opinion, adding Table 3 after Section 3.2 to display the differential metabolites relative to the CK group in each group. We hope we have not misunderstood your suggestion; if you believe more details are needed, we might further supplement them in the supplemental material.

Comment8: The introduction requires rewriting for better coherence, as it currently reads more like a literature review. For example, the sentence “This introduction outlines the…” could be restructured for smoother flow.
Response8: Agreed. We have attempted to reorganize the introduction, and you can see that its structure and content have undergone some changes. Overall, following your suggestion, we believe the current version is indeed much smoother, thank you very much!

Comment9: In some places Bacillus are italicised while in some places not.
Response9: We believe this is another regrettable oversight. We have checked the entire document, and the new version should have completely fixed this issue.

Round 2

Reviewer 1 Report

Comments and Suggestions for Authors

The authors have revised the manuscript and made some edits. However, the article as presented is not any better. There are many inaccuracies in the presentation of the experimental material, the name of the object (different abbreviations are used), and the captions of figures and tables).  Some comments and suggestions are presented in the text.

Comments on the Quality of English Language

The presentation of the text in English is not always satisfactory. It should be corrected.

Author Response

Comment 1: To clarify: biosynthesis of phenolic compounds includes the phenylpropanoid and flavonoid pathways. In this regard, the text needs to be corrected.

Response 1: Agree. We tried rewrite the sentence as shown at line 25-28. We hope that the new sentence faithfully conveys the facts.

Comment 2:The first sentence in this section corresponds to the data obtained. The second requires correction, it is not for the abstract in its presented form.

Response 2: We are not entirely certain if we correctly understood the reviewer's comment on the last sentence of the abstract, but we have attempted to replace the original evaluative language with a more pragmatic sentence.

Comment 3:This issue has not been studied. Only changes in the formation of metabolites during infection have been analyzed.

Response 3: We understand your concerns. It seems that the sentence may have been too concise, leading to potential ambiguity due to the omission of critical information. Thus, we have reviewed the descriptions of soil microorganisms, material exchange and related concepts within the paragraph, revised the original sentence to be more comprehensive. Now you can find the new sentence at lines 48-51. We hope that we have not misinterpreted your intentions.

Comment 4:The pine needle samples were rinsed with sterile water and then subjected to vacuum 174 freeze-drying. An extraction solvent (a mixture of methanol, acetonitrile, and water in a 175 volume ratio of 2:2:1) was added and mixed evenly. Is “acetonitrile” right for here?

Response 4: We confirm that there are no inaccuracies in the description of the technical method; the extraction solvent indeed consists of methanol, acetonitrile, and water mixed in the specified proportions.

Comment 5:Same with 4: a mixture of acetonitrile and water in a volume ratio of 1:1, is “acetonitrile” right for here?

Response 5: We confirm that there are no inaccuracies in the description of the technical method; the extraction solvent indeed consists of methanol, acetonitrile, and water mixed in the specified proportions.

Comment 6:Photos must be presented in the same format (either growing in pots or taken out of them). And taken from the same angle.

Response 6: Agree. Due to issues with personnel and storage media, we regret to inform that some images have been lost. We have endeavored to provide photographs that faithfully document the experimental progress, though they may not meet the desired standards. We recognize and appreciate your stringent requirements for image quality. However, it appears that supplementing and revising these images within a short timeframe is not feasible. Nonetheless, we are grateful for your meticulous attention to detail.

Comment 7:We analyzed differentially expressed metabolites within KEGG pathways across treatment groups after three days (Figure 3). What is that mean of “three days”?

Response 7: Thank you for pointing out the ambiguity in that sentence. We have revised the phrasing, and you can now find the new sentence at lines 235-237. Correspondingly, we have also modified the description in the caption of Figure 3.

Comment 8:Csufcsp75 induction notably increased activity in amino acid metabolism pathways, particularly pathways involved in amino acid degradation and synthesis. Edit text.

Response 8: Agree. The original statement did not convey sufficient valid information. We have attempted to rephrase the sentence, which can now be found at lines 242-245. We believe the new phrasing faithfully reflects the original intent and hope that it is now sufficiently accurate.

Reviewer 3 Report

Comments and Suggestions for Authors

In this revised version, the authors have provided comprehensive details on the experimental design, methods, and accession numbers, making the manuscript clear, easy to follow, and well-suited for publication.

Author Response

Comment:In this revised version, the authors have provided comprehensive details on the experimental design, methods, and accession numbers, making the manuscript clear, easy to follow, and well-suited for publication.

Response:  We are extremely grateful for the reviewer's efforts to enhance this manuscript, and we appreciate the affirmation and encouragement! Thank you so much for your work in ensuring the rigorous and fair publication of scientific outcomes!